# Design Considerations for the Liquid Air Energy Storage System Integrated to Nuclear Steam Cycle

**Seok-Ho Song, Jin-Young Heo and Jeong-Ik Lee \***

Department of Nuclear and Quantum Engineering, Korea Advanced Institute of Science and Technology, Daejeon 305-701, Korea; 1812wow@kaist.ac.kr (S.-H.S.); jyh9090@kaist.ac.kr (J.-Y.H.)
\* Correspondence: jeongiklee@kaist.ac.kr

**Abstract:** A nuclear power plant is one of the power sources that shares a large portion of base-load. However, as the proportion of renewable energy increases, nuclear power plants will be required to generate power more flexibly due to the intermittency of the renewable energy sources. This paper reviews a layout thermally integrating the liquid air energy storage system with a nuclear power plant. To evaluate the performance realistically while optimizing the layout, operating nuclear power plant conditions are used. After revisiting the analysis, the optimized performance of the proposed system is predicted to achieve 59.96% of the round-trip efficiency. However, it is further shown that external environmental conditions could deteriorate the performance. For the design of liquid air energy storage-nuclear power plant integrated systems, both the steam properties of the linked plants and external factors should be considered.

**Keywords:** liquid air energy storage (LAES); nuclear power; load following operation

## 1. Introduction

The concerns around global warming and environmental issues are growing, and thus the development of eco-friendly renewable energies plays an important role as a solution [1–3]. Unfortunately, renewable energy sources such as solar and wind, unlike conventional power generation systems, cannot produce electricity consistently and follow the demand. It means that renewable power sources cannot generate extra energy or reduce output power due to sudden changes in demand [4]. They should, therefore, have an energy storage system that can store a large amount of energy and discharge when needed. To solve this problem of reliability, research has been conducted to integrate renewable sources with the energy storage system (ESS). The integrated ESS supports renewable power systems by playing the function of time shifting or moving electric load from one time to another. ESS can also smooth out the sharp output from renewable power sources that could damage the electricity grid [5–7].

Furthermore, to optimize renewable power sources, attention should be focused on electricity generation as well. As renewable power sources such as wind and solar have a high dependency on time and location, they cannot be controlled artificially. To maximize the efficiency of renewable energy and to reduce loss from the use of renewable energy, balancing the existing electricity supply and renewable energy supply is essential. According to research from the National Renewable Energy Lab (NREL), the production of electricity from solar power can affect the supply of electricity from conventional sources. According to the data provided from the California grid [8], the increase of solar power leads to the decrease of remaining net load during the daytime and increase at a rapid rate after the sun sets. Known as the duck curve phenomenon, this effect occurs because the load needs to balance out the overgeneration for the grid stability [9]. As the renewable sources, mainly solar and wind, take a greater portion in the grid, the energy sources that have been generated at a stable base load are required to change their load during the day.

The intermittency problem of renewable energy can be alleviated with a flexible operation of conventional power plants such nuclear and coal power. In this study, the authors focus on the concept of directly coupling ESS to the conventional power plant to enhance flexibility. However, there is also a study suggesting that renewable energy with ESS can support grid stabilization when a nuclear or coal power plant goes out of service unexpectedly [10]. Thus, the grid stability can be improved by adding ESS to either conventional energy sources or renewable energy sources.

Nuclear power is one of the main sources of baseload generation in many countries [11]. As previously mentioned, the increase of renewable sources inevitably demands the load reduction of existing nuclear plants. Although some nuclear power plants have the capacity to control the output power following the demand (load following), not all of them can be operated in load following mode. Even if they do so, the periodic output changes in the core can affect the service time of safety important equipment and can even deteriorate the integrity of the reactor core [12]. Such load reduction strategies can also curtail the economic return from the nuclear plants [13], since they economically perform best when operating at full load to maximize the generation profit from the large initial investment.

An alternative way of shifting the load of conventional nuclear plants is to integrate a large-scale ESS to the secondary system of a nuclear power plant; a system from steam generator to steam turbine. Without changing the reactor output, this method stores energy available from the steam cycle during the period of overgeneration and generates extra energy during high demand. The solution is a better way to retrofit nuclear power plants that were previously unable to follow the load due to stability problems. However, the ESS technologies suitable for such integration need to be scalable to the level of nuclear plants and affordable at such sizes.

One of the potential candidates is the liquid air energy storage (LAES) system, recently receiving attention due to its potential for fast deployment [14]. A research team from the University of Birmingham suggested a conceptual study combining a liquid air energy storage system with a light water reactor [15]. Their results showed that the round-trip efficiency would reach approximately 71%, and output work would be approximately 2.7 times higher than that of the existing nuclear power plant alone. However, the suggested design poses several technical challenges, as marked in Figure 1. First, the discharge mode involves 100% steam bypass from the nuclear secondary side which poses integrity issues to the nuclear power plant's steam cycle. Second, excessively ideal design conditions were used for the performance prediction, including zero pressure drop and low pinch temperatures across heat exchangers [16]. Such assumptions are not suitable for a technically feasible design, especially when nuclear retrofitting requires more realistic and safety-based considerations.

The objective of this paper is to reevaluate the integration of LAES with existing nuclear plants under more realistic design conditions. The research analyzes the cycle layout presented by the reference [15] using the developed in-house code and compares the results with those from the reference paper. Then, the design conditions including pinch temperature and pressure drop are modified to be more realistic, and operating conditions from the currently operating nuclear power plant APR1400 in South Korea are utilized for more practical integration of the LAES system. The research highlights whether the integration design can be considered technically feasible for future adoption in the operating nuclear power plant. It is further noted that the APR1400 shares common characteristics with the other Generation III nuclear power plants, such as AP1000 of Westinghouse and EPR of formerly AREVA. Therefore, the conclusion of this study is not limited to just one specific nuclear power plant design.

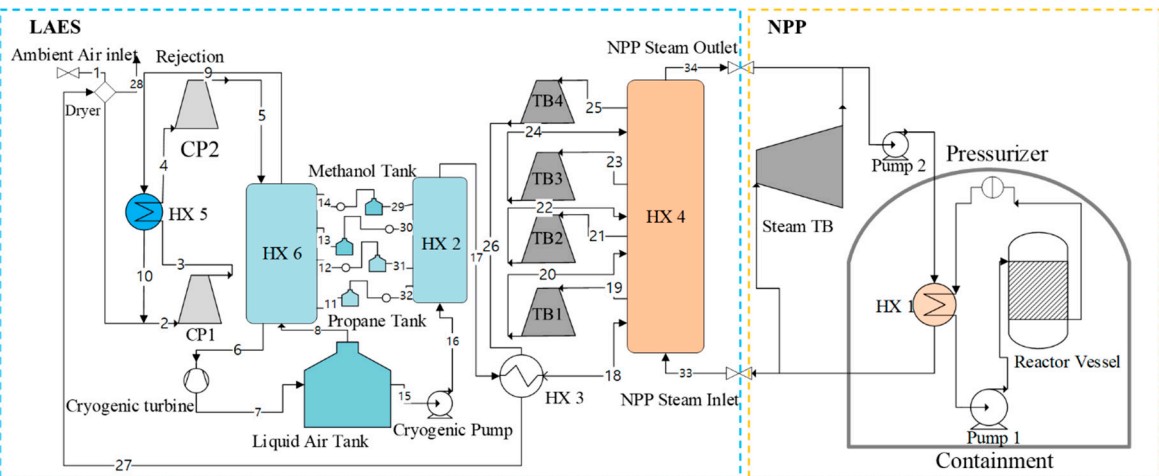

**Figure 1.** LAES system with a reference nuclear power plant (NPP) system with design limitations adapted from [15].

## 2. Analysis Methodology

### 2.1. General Code

The first goal of this research is to reproduce the model proposed in the reference [15] with the KAIST Closed Cycle Design code (KAIST-CCD). This work gives an insight to how the LAES system performs and how each design parameter will influence the round-trip efficiency. After the model is verified to produce similar results to the previous study, the model can be modified to optimize the operating conditions of the LAES system with APR1400 under more realistic conditions.

To test the LAES system under realistic conditions, three design parameters were modified from the reference [15].

(1) Pinch temperature of heat exchanger: The pinch temperature for heat exchangers in the reference [15] is 2K pinch. However, when designing a heat exchanger, typically 5K or larger values are assumed for the pinch due to economic reason and increased pressure drop for excessively low pinch design condition. Therefore, in this paper, the pinch temperature in the heat exchangers is changed to 5K.

(2) Pressure drop: Pressure drop in components and piping always occurs. Due to this pressure drop, the pressure of fluid decreases while flowing through the system. This generally influences the overall performance of the system.

(3) Ambient temperature: In the realistic system, the ambient temperature is always changing. As an LAES system is an open system that takes in air from the atmosphere, the temperature of ambient air affects the performance of the system.

The KAIST-CCD is a MATLAB based code used to calculate and optimize thermodynamic processes. It can be separated into two main subroutines: the layout section and the design parameter section. The layout section determines the process layout of a thermal system, and it gives the information on how components are connected to each other. In the design parameter section, it provides the design conditions for the layout code: the pressure and temperature for ambient air, the efficiency, and operating conditions for each component. The properties of fluid are obtained from REFPROP developed by NIST [17].

The layout adopted for code modelling is depicted in Figure 1. It is separated into two parts: before the liquid air tank and after the liquid air tank. The layout before the liquid air tank is the storage part of the system (i.e., charging mode) and the layout after the liquid air tank is the system for discharge operation. The code calculates the charging system first, until the result converges in the charging system and then starts to calculate the discharging system with the results from the charging system. The algorithm of the code is shown in Figure 2. The following describes the thermodynamic for constructing the model of the proposed system. The calculation in the components follows the fundamental laws

of general thermodynamics. However, certain components follow the calculations below for better accuracy. The specific parameters for calculation are stated in Tables 1 and 2.

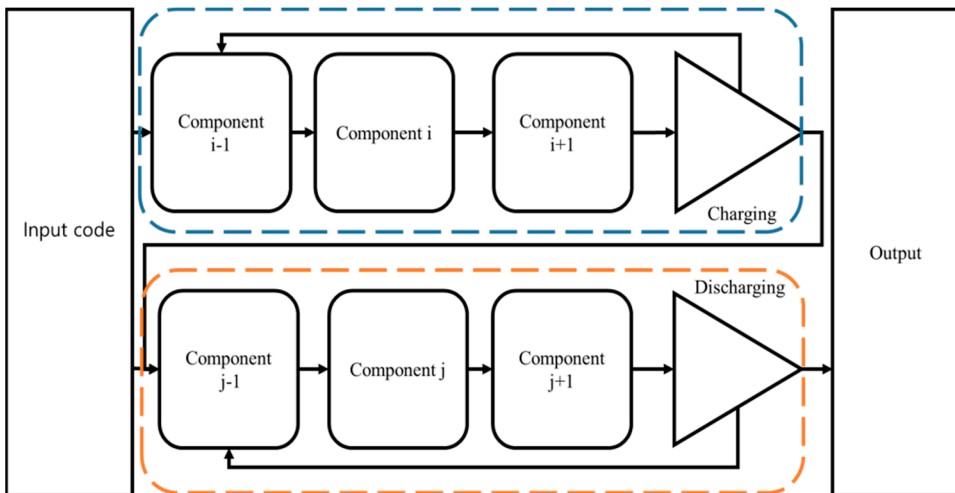

**Figure 2.** Overall algorithm structure of the KAIST-CCD code.

**Table 1.** Parameters for LAES Calculation data from [15].

| Variables | Values |
|---|---|
| Thermal power from NPP (MW) | 250 |
| NPP inlet steam pressure (kPa) | 7093 |
| NPP inlet steam temperature (K) | 560 |
| Thermal efficiency of the NPP (%) | 31 |
| Ambient pressure (kPa) | 101 |
| Ambient temperature (K) | 288 |
| Liquid air storage pressure (kPa) | 101 |
| Operational period in energy storage mode (hours/day) | 8 |
| Operational period in energy release mode (hours/day) | 1 |
| Temperature approach of heat exchangers (K) | 2 |
| Isentropic efficiency of the air turbines (%) | 92 |
| Isentropic efficiency of the cryogenic turbine (%) | 88 |
| Isothermal efficiency of air compressors (%) | 90 |
| Isentropic efficiency of the cryogenic pump (%) | 70 |

**Table 2.** Nuclear Power Plant Conditions—APR1400.

| Variables | Values |
|---|---|
| Thermal power from NPP (MW) | variable |
| NPP inlet steam pressure (kPa) | 1352 |
| NPP inlet steam temperature (K) | 500 |
| Thermal efficiency of the NPP (%) | 31 |
| Ambient pressure (kPa) | 101 |
| Ambient temperature (K) | 288 |
| Liquid air storage pressure (kPa) | 101 |
| Operational period in energy storage mode (hours/day) | 8 |
| Operational period in energy release mode (hours/day) | 1 |
| Temperature approach of heat exchangers (K) | 5 |
| Isentropic efficiency of the air turbines (%) | 92 |
| Isentropic efficiency of the cryogenic turbine (%) | 88 |
| Isothermal efficiency of air compressors (%) | 90 |
| Isentropic efficiency of the cryogenic pump (%) | 70 |

### 2.2. Compressor (Isothermal)

In Figure 1, for flow numbers 2 to 3 and 4 to 5, two isothermal compressors were used for the system in the reference [15]. In paper [15], the exact method of calculation for an isothermal compressor was not presented. Therefore, the analysis methodology in this paper follows the reference [18], which presents how to evaluate an isothermal compressor. In the isothermal compressor model, inlet temperature and pressure, target pressure, and the compression process number are required for the performance evaluation. The compression process number determines the numerical discretization of a continuous process to evaluate work and cooling in the isothermal compressor. As shown in Figure 3, in order to describe isothermal compression, compression and cooling are repeated several times numerically. In this paper, the step number (i.e., numerical discretization) of the isothermal compressors is set to 500, a large enough number to reproduce the outcome of reference [15]. The final compression work required, which is shown in Equation (1), is calculated by summing up the discretized work provided through the whole process.

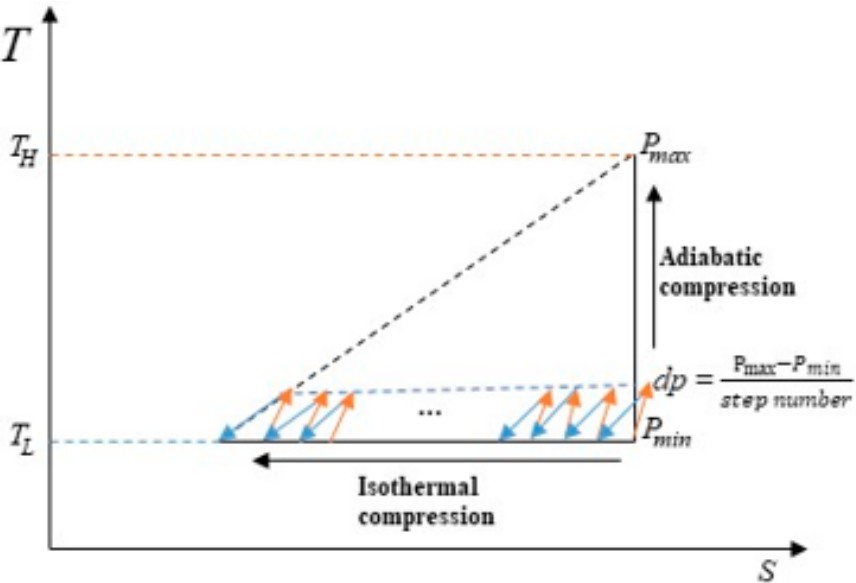

**Figure 3.** T-s diagram of the isothermal compression adapted from [18].

$$W_{cp} = \Sigma_{i=1}^{n} W_{CP,i} = \dot{m} \Sigma_{i=1}^{n} \frac{(h_{CP,i} - h_{CP,i-1})}{\eta_{CP}} \tag{1}$$

$$n = \text{step number}$$

$$h_{CP,0} = h_{CP,inlet}$$

### 2.3. Heat Exchanger

The methodology is applied for evaluating all heat exchangers except heat exchanger 4 in Figure 1. The heat exchanger model requires the following information: the inlet temperature and pressure of the hot and cold side, the pinch temperature, and the pressure drop condition.

The inlet temperature and pressure of the hot and cold side, pinch temperature, and pressure drop condition are required for the heat exchanger calculation. As heat exchangers include 3-way heat exchangers, the effectiveness model is used for the calculation:

$$\epsilon = \frac{Q_{actual}}{Q_{max}} = \frac{\left( \dot{m}_{cold1} \left( h_{cold1,out} - h_{cold1,in} \right) + \dot{m}_{cold2} \left( h_{cold2,out} - h_{cold2,in} \right) \right)}{\dot{m}_{hot} \left( h_{hot,in} - h_{hot,out,ideal} \right)}$$
$$= \frac{h_{hot,in} - h_{hot,out}}{h_{hot,in} - h_{hot,out,ideal}}, \ \left( h_{hot,out,ideal} = h(\max(T_{cold1,in}, T_{cold2,in})) \right) \tag{2}$$

Effectiveness is assumed to obtain $h_{hot,out}$. If the calculated pinch is lower than the prescribed condition, then the effectiveness value is decreased until the calculated pinch exceeds the prescribed pinch conditions. The process is shown in Figure 4.

$$h_{hot,out} = h_{hot,in} - \epsilon \cdot (h_{hot,in} - h_{hot,out,ideal}) \tag{3}$$

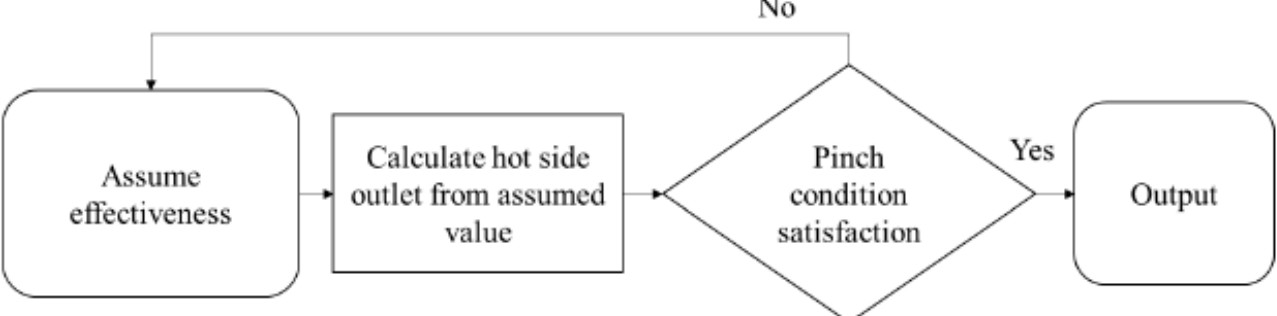

**Figure 4.** Effectiveness model algorithm.

### 2.4. Heat Exchanger (Air to Steam)

As shown in Figure 1, heat exchanger 4 is designed to exchange heat between air and steam flow channels. This model requires information on the mass flow rate of air, the inlet temperature and pressure of air side and steam side, the outlet temperature of steam side, and pinch and pressure drop conditions. As the temperature of the steam side is fixed to integrate with the steam cycle of the APR1400, the pinch condition should be satisfied by changing the air side temperature.

Figure 5 shows that the initial calculation of air temperature profile may result in an overlap with the steam temperature profile. To avoid the overlap, the cold side outlet temperature is corrected in the first iteration until a result satisfying the pinch condition is obtained. Finally, the mass flow rate of steam side is calculated as shown in Equation (4):

$$\dot{m}_{steam} = \frac{\dot{m}_{air}(h_{air,out} - h_{air,in})}{h_{steam,in} - h_{steam,out}} \tag{4}$$

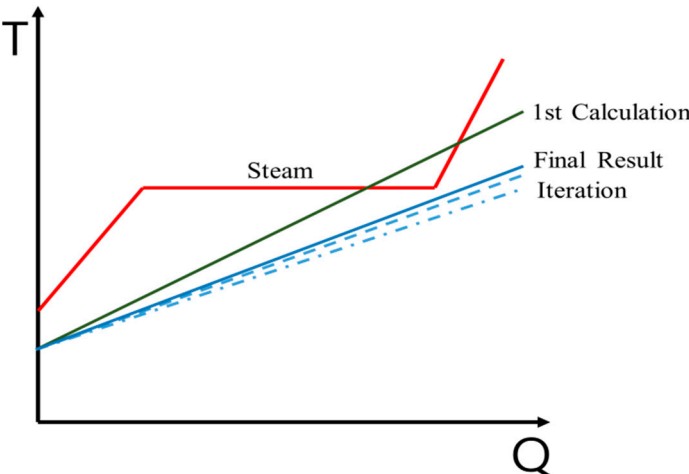

**Figure 5.** Temperature (T)–heat load (Q) diagram of heat exchanger 4.

*2.5. Exergy Calculation*

From the input code, the temperature and pressure of ambient conditions are used to calculate ambient enthalpy and ambient entropy for exergy calculations. The exergy for each point is calculated as follows [19]:

$$e_i = h_i - h_{amb} - T_{amb}(s_i - s_{amb}) \tag{5}$$

The exergy destruction calculation of the components is performed next:

$$X = \Sigma \dot{m}_{in} \cdot e_{in} - \Sigma \dot{m}_{out} \cdot e_{out} + \Sigma W \tag{6}$$

*2.6. Round-Trip Efficiency*

Round-trip efficiency is the most important parameter to evaluate the performance of energy storage systems. It is the ratio of energy released by the system and energy consumed by the system. In the case of LAES integrated NPP, the power from NPP is reduced during the energy release. This is because the system utilizes power from the NPP through steam bypass while operating in discharge mode. Moreover, the system is designed to have storage mode eight times longer than release mode. This is also considered in the calculation of round-trip efficiency.

The total power from release mode in LAES system is a sum of power from air turbines and power consumption of cryogenic pump. This process is shown in the Figure 1, flow number 15 to 32.

$$W_{ER} = \Sigma W_{air\ TB} - W_{cryogenic\ pump} \tag{7}$$

The power loss from NPP is calculated from the efficiency of NPP, steam flow rate, and enthalpy from inlet and outlet of NPP steam side. This is number 33 and 34 in Figure 1. The specific values of efficiency and boundary conditions for NPP are in Tables 1 and 2, respectively.

$$W_{NPP} = \eta_{NPP} \cdot m_{steam} \cdot (h_{steam_{in}} - h_{steam_{out}}) \tag{8}$$

The power consumption from storage mode in LAES system is sum of power consumption from compressors and power output from cryogenic turbine. This process is described in the Figure 1, flow number 1 to 14.

$$W_{ES} = \Sigma W_{CP} - W_{cryogenic\ TB} \tag{9}$$

The round-trip efficiency of LAES system is calculated by the total power from release mode, power loss from NPP, and power consumption from storage mode following Equation (9). The operation time for each mode is shown in Tables 1 and 2.

$$\eta_{RTE} = \frac{(W_{ER} - W_{NPP}) \cdot t_{ER}}{W_{ES} \cdot t_{ES}} \tag{10}$$

## 3. Result and Discussion

*3.1. Comparison of Result*

When all the parameters are assumed to be the same as the reference [15], the results from the reference and the results obtained from this study are compared first to verify KAIST-CCD for investigating this problem. There are differences between steam parameters obtained in this study compared to the values reported in the reference [15].

The most significant difference is the mass flow change from point 9 to 10. This is due to the difference in yield for producing liquid air. The reference [15] shows 84% liquid air yield; therefore, it has less amount of air flowing in stations 9 and 10. However, the current study shows 83% liquid air yield, and this difference increases the amount of gaseous air in the liquid air tank, and the mass flow rate in station 9 is also increased as a result. Moreover, this effect induces the temperature difference in stations 3 to 5 as it is connected with stations 9 and 10 via a heat exchanger.

This difference in liquid air yield is due to the difference of temperature in the flow number 7 (the inlet of liquid air tank), which seems to be rooted in two reasons. The first cause is the difference in the thermal properties database. The thermal properties used in this study are from REFROP 10.0 while the reference [15] used the older version, REFROP 8.0. As shown in Table 3, station numbers 7 and 15, the reference shows the temperature of vapor air to be 81K (flow number 7), and the temperature of liquid air as 80K (flow number 15). In the current study, the temperatures of vapor air and liquid air are both 79K due to the difference in the fluid thermal properties database. The second cause is the difference in how isothermal compressor is modeled between the reference [15] and the current study. Since not enough detail is provided in reference [15] for modeling the isothermal compressor, this study adopted the approach presented in reference [18] for the isothermal compressor model. As a result, the outlet temperature of the isothermal compressor obtained in this study is different with the reference [15].

**Table 3.** The main steam parameter for reference data from [15] (left) and KAIST-CCD (right).

| | Reference [15] | | | KAIST-CCD | | | |
|---|---|---|---|---|---|---|---|
| Flow No. | Mass Flow (kg/s) | Pressure (kPa) | Temperature (K) | Mass Flow (kg/s) | Pressure (kPa) | Temperature (K) | Fluid Type |
| 1 | 150 | 101 | 288 | | Ambient air intake | | |
| 2 | 179 | 101 | 288 | 179 | 101 | 288 | Air |
| 3 | 179 | 1159 | 288 | 179 | 1159 | 289 | Air |
| 4 | 179 | 1159 | 282 | 179 | 1159 | 287 | Air |
| 5 | 179 | 13,409 | 288 | 179 | 13,409 | 289 | Air |
| 6 | 179 | 13,409 | 102 | 179 | 13,409 | 104 | Air |
| 7 | 179 | 101 | 81 | 179 | 101 | 79 | Air |
| 8 | 29 | 101 | 83 | 33 | 101 | 82 | Air |
| 9 | 29 | 101 | 250 | 33 | 101 | 250 | Air |
| 10 | 29 | 101 | 288 | 33 | 101 | 287 | Air |
| 11 | 167 | 101 | 95 | 167 | 101 | 95 | Propane |
| 12 | 167 | 101 | 212 | 167 | 101 | 212 | Propane |
| 13 | 90 | 101 | 219 | 90 | 101 | 219 | Methanol |
| 14 | 90 | 101 | 286 | 90 | 101 | 286 | Methanol |
| 15 | 1195 | 101 | 80 | 1170 | 101 | 79 | Air |
| 16 | 1195 | 11,385 | 83 | 1170 | 11,385 | 85 | Air |
| 17 | 1195 | 11,385 | 283 | 1170 | 11,385 | 286 | Air |
| 18 | 1195 | 11,385 | 380 | 1170 | 11,385 | 405 | Air |
| 19 | 1195 | 11,385 | 553 | 1170 | 3497 | 558 | Air |
| 20 | 1195 | 3497 | 396 | 1170 | 3497 | 412 | Air |
| 21 | 1195 | 3497 | 553 | 1170 | 3497 | 558 | Air |
| 22 | 1195 | 1074 | 397 | 1170 | 1074 | 413 | Air |
| 23 | 1195 | 1074 | 553 | 1170 | 1074 | 558 | Air |
| 24 | 1195 | 330 | 397 | 1170 | 330 | 414 | Air |
| 25 | 1195 | 330 | 553 | 1170 | 330 | 558 | Air |
| 26 | 1195 | 101 | 397 | 1170 | 110 | 422 | Air |
| 27 | 1195 | 101 | 288 | 1170 | 110 | 288 | Air |
| 28 | 1195 | 101 | 288 | | Rejection | | |
| 29 | 723 | 101 | 288 | 723 | 101 | 288 | Methanol |
| 30 | 723 | 101 | 217 | 723 | 101 | 217 | Methanol |
| 31 | 1337 | 101 | 214 | 1337 | 101 | 214 | Propane |
| 32 | 1337 | 101 | 93 | 1337 | 101 | 93 | Propane |
| 33 | 442 | 7093 | 560 | 442 | 7093 | 560 | Water |
| 34 | 442 | 7093 | 493 | 442 | 7093 | 493 | Water |

For the performance of components, KAIST-CCD results show similar power consumption and generation with the values reported in the reference [15]. As shown in Table 4, the difference exists in the exergy destruction in a cryogenic turbine and a cryogenic pump. These are the components that deal with the cryogenic air modeled as the mixture of several gas components. In the cryogenic region, especially in a saturated liquid state of air, a small difference in temperature of 0.5 to 1K could result in a large difference in enthalpy and entropy. The difference in the exergy loss for cryogenic components is due to these characteristics.

**Table 4.** Performance summary for reference data from [15] (left) and KAIST-CCD (right).

| | Reference [15] | | KAIST-CCD | |
|---|---|---|---|---|
| | **Power (MW)** | **Exergy Loss (MW)** | **Power (MW)** | **Exergy Loss (MW)** |
| | *Energy storage mode* | | | |
| Compressor 1 | 40.03 | 3.96 | 40.11 | 4.05 |
| Compressor 2 | 39.83 | 3.99 | 39.65 | 3.82 |
| Cryogenic turbine | 3.12 | 0.43 | 3.32 | 1.64 |
| | *Energy release mode* | | | |
| Cryogenic pump | 19.18 | 5.77 | 21.35 | 22.15 |
| Air turbines | 706.69 | 62.81 | 689.00 | 42.22 |
| Net power consumption in Storage mode (MW) | | 76.74 | | 76.45 |
| Net power output in release mode (MW) | | 687.51 | | 667.65 |
| Round-trip efficiency (%) | | 71.26 | | 68.29 |

## 3.2. Pinch Effect

As the increasing pinch is assumed to utilize a less effective heat exchanger, it is natural to expect that the liquid air yield and the round-trip efficiency of the system will be reduced compared to the previous 2K pinch assumption in the reference. The temperature of the coolant follows Table 5. The outlet temperature of methanol from the heat exchanger 6 changes with the assumed pinch temperatures. For instance, the outlet temperature of methanol (i.e., cold side) from the heat exchanger 6 is set to 283K since the hot side (i.e., air side) inlet is 288K while the pinch temperature is assumed to be 5K.

**Table 5.** Temperature of propane and methanol.

| Pinch (K) | 2 | 5 |
|---|---|---|
| Inlet Temperature for Propane (K) | 95 | 95 |
| Outlet Temperature for Propane (K) | 212 | 212 |
| Inlet Temperature for Methanol (K) | 219 | 219 |
| Outlet Temperature for Methanol (K) | 286 | 283 |

As expected, the round-trip effectiveness was lower for 5K pinch compared to 2K pinch. As shown in Table 6, In the storage mode, the case of 2K pinch consumed less power than the case of 5K pinch. In the release mode, more energy was produced in the 2K pinch case than in the 5K pinch case. It led to a similar trend for the difference in power output. As shown in Table 7, the inlet temperatures for air turbine in 5K pinch case and for 2K pinch case are 555K and 558K, respectively. These differences resulted in a round-trip efficiency difference of about 1%.

## 3.3. Realistic Nuclear Power Plant Model (APR1400)

Considering the realistic integration with NPP and LAES system, the steam from NPP cannot be bypassed 100% to the LAES discharging cycle. Operating between full power and complete shutdown of the secondary side may cause substantial thermal stress to the turbine components, and this operation method requires more time for hot startup [20]. Therefore, this paper suggests a more realistic integration to the reference nuclear plant, APR1400. The portion of steam bypass is modified to 20% of the total mass flow rate in the secondary side to avoid the aforementioned operational issues. Another modification is the steam condition of the flow sent to the steam-air heat exchanger (heat exchanger 4), as this work assumes the superheated steam to be split from the inlet of the low-pressure turbine, which the temperature and pressure are shown in Table 2.

**Table 6.** Performance summary for 5K pinch (left) and 2K pinch (right).

| | 5K Pinch | | 2K Pinch | |
|---|---|---|---|---|
| | **Power (MW)** | **Exergy Loss (MW)** | **Power (MW)** | **Exergy Loss (MW)** |
| | | Energy storage mode | | |
| Compressor 1 | 40.11 | 4.05 | 40.11 | 4.05 |
| Compressor 2 | 39.87 | 4.03 | 39.65 | 3.82 |
| Cryo-turbine | 3.30 | 1.64 | 3.32 | 1.64 |
| | | Energy release mode | | |
| Cryogenic pump | 21.38 | 22.18 | 21.35 | 22.15 |
| Air turbines | 686.19 | 42.28 | 689.00 | 42.22 |
| LAES Yield | | 0.81793 | | 0.81676 |
| Net power consumption in Storage mode (MW) | | 76.68 | | 76.45 |
| Net power output in release mode (MW) | | 664.81 | | 667.65 |
| Round-trip efficiency (%) | | 67.62 | | 68.29 |

**Table 7.** The main steam parameter for 5K pinch (left) and 2K pinch (right).

| | 5K Pinch Case | | | 2K Pinch Case | | | |
|---|---|---|---|---|---|---|---|
| **Flow No.** | **Mass Flow (kg/s)** | **Pressure (kPa)** | **Temperature (K)** | **Mass Flow (kg/s)** | **Pressure (kPa)** | **Temperature (K)** | **Fluid Type** |
| 1 | | | | Inhalation | | | |
| 2 | 179 | 101 | 288 | 179 | 101 | 288 | Air |
| 3 | 179 | 1159 | 289 | 179 | 1159 | 289 | Air |
| 4 | 179 | 1159 | 288 | 179 | 1159 | 287 | Air |
| 5 | 179 | 13,409 | 290 | 179 | 13,409 | 289 | Air |
| 6 | 179 | 13,409 | 104 | 179 | 13,409 | 104 | Air |
| 7 | 179 | 101 | 79 | 179 | 101 | 79 | Air |
| 8 | 33 | 101 | 82 | 33 | 101 | 82 | Air |
| 9 | 33 | 101 | 268 | 33 | 101 | 250 | Air |
| 10 | 33 | 101 | 284 | 33 | 101 | 287 | Air |
| 11 | 167 | 101 | 95 | 167 | 101 | 95 | Propane |
| 12 | 167 | 101 | 212 | 167 | 101 | 212 | Propane |
| 13 | 90 | 101 | 219 | 90 | 101 | 219 | Methanol |
| 14 | 90 | 101 | 283 | 90 | 101 | 286 | Methanol |
| 15 | 1171 | 101 | 79 | 1170 | 101 | 79 | Air |
| 16 | 1171 | 11,385 | 85 | 1170 | 11,385 | 85 | Air |
| 17 | 1171 | 11,385 | 282 | 1170 | 11,385 | 286 | Air |
| 18 | 1171 | 11,385 | 400 | 1170 | 11,385 | 405 | Air |
| 19 | 1171 | 11,385 | 555 | 1170 | 11,385 | 558 | Air |
| 20 | 1171 | 3497 | 410 | 1170 | 3497 | 412 | Air |
| 21 | 1171 | 3497 | 555 | 1170 | 3497 | 558 | Air |
| 22 | 1171 | 1074 | 411 | 1170 | 1074 | 413 | Air |
| 23 | 1171 | 1074 | 555 | 1170 | 1074 | 558 | Air |
| 24 | 1171 | 330 | 411 | 1170 | 330 | 414 | Air |
| 25 | 1171 | 330 | 555 | 1170 | 330 | 558 | Air |
| 26 | 1171 | 110 | 420 | 1170 | 110 | 422 | Air |
| 27 | 1171 | 110 | 287 | 1170 | 110 | 288 | Air |
| 28 | | | | Rejection | | | |
| 29 | 723 | 101 | 288 | 723 | 101 | 288 | Methanol |
| 30 | 723 | 101 | 217 | 723 | 101 | 217 | Methanol |
| 31 | 1337 | 101 | 214 | 1337 | 101 | 214 | Propane |
| 32 | 1337 | 101 | 93 | 1337 | 101 | 93 | Propane |
| 33 | 412 | 7093 | 560 | 442 | 7093 | 560 | Water |
| 34 | 412 | 7093 | 493 | 442 | 7093 | 493 | Water |

Under these conditions, the temperature profile inside the heat exchanger 4 creates a unique design issue due to the condensation of steam occurring at the steam side. The NPP side steam changes phase from a superheated condition to a subcooled condition, as shown in Figure 6. The heat exchange between air and steam in LAES in the discharging cycle leads to a significant difference between the air and steam temperature profiles. Hence, the heat exchanger has the highest exergy destruction in release mode, as shown in Figure 7. However, the temperature profile has the advantage of maximizing latent heat from the steam side. Comparing point 33 in Tables 6 and 8, the APR1400 condition allows the LAES system to operate with a relatively low flow rate. In the condition of APR1400, since the entry temperature of steam is low, the entry temperature of the air turbine is also low. As shown in Table 8, the turbine inlet temperature of the LAES discharging cycle is 471K, which is about 87K lower than the turbine inlet temperature from Table 7. The turbine inlet temperature difference resulted in relatively low power output, as shown in Table 9. However, in actual nuclear power plant conditions, it is no longer possible to link LAES with high temperature, so the practical limit of round-trip efficiency is expected to be about 60% at the maximum.

**Table 8.** The main steam parameter for APR1400 integrated LAES.

| Flow No. | Mass Flow (kg/s) | Pressure (kPa) | Temperature (K) | Fluid |
|---|---|---|---|---|
| 1 | | | Inhalation | |
| 2 | 179 | 101 | 288 | Air |
| 3 | 179 | 1159 | 289 | Air |
| 4 | 179 | 1159 | 288 | Air |
| 5 | 179 | 13,409 | 290 | Air |
| 6 | 179 | 13,409 | 104 | Air |
| 7 | 179 | 101 | 79 | Air |
| 8 | 33 | 101 | 82 | Air |
| 9 | 33 | 101 | 268 | Air |
| 10 | 33 | 101 | 284 | Air |
| 11 | 167 | 101 | 95 | Propane |
| 12 | 167 | 101 | 212 | Propane |
| 13 | 90 | 101 | 219 | Methanol |
| 14 | 90 | 101 | 283 | Methanol |
| 15 | 1171 | 101 | 79 | Air |
| 16 | 1171 | 11,385 | 85 | Air |
| 17 | 1171 | 11,385 | 282 | Air |
| 18 | 1171 | 11,385 | 341 | Air |
| 19 | 1171 | 11,385 | 472 | Air |
| 20 | 1171 | 3497 | 346 | Air |
| 21 | 1171 | 3497 | 471 | Air |
| 22 | 1171 | 1074 | 347 | Air |
| 23 | 1171 | 1074 | 471 | Air |
| 24 | 1171 | 330 | 348 | Air |
| 25 | 1171 | 330 | 471 | Air |
| 26 | 1171 | 110 | 356 | Air |
| 27 | 1171 | 110 | 287 | Air |
| 28 | | | Rejection | |
| 29 | 723 | 101 | 288 | Methanol |
| 30 | 723 | 101 | 217 | Methanol |
| 31 | 1337 | 101 | 214 | Propane |
| 32 | 1337 | 101 | 93 | Propane |
| 33 | 270 | 1458 | 500 | Water |
| 34 | 270 | 1346 | 412 | Water |

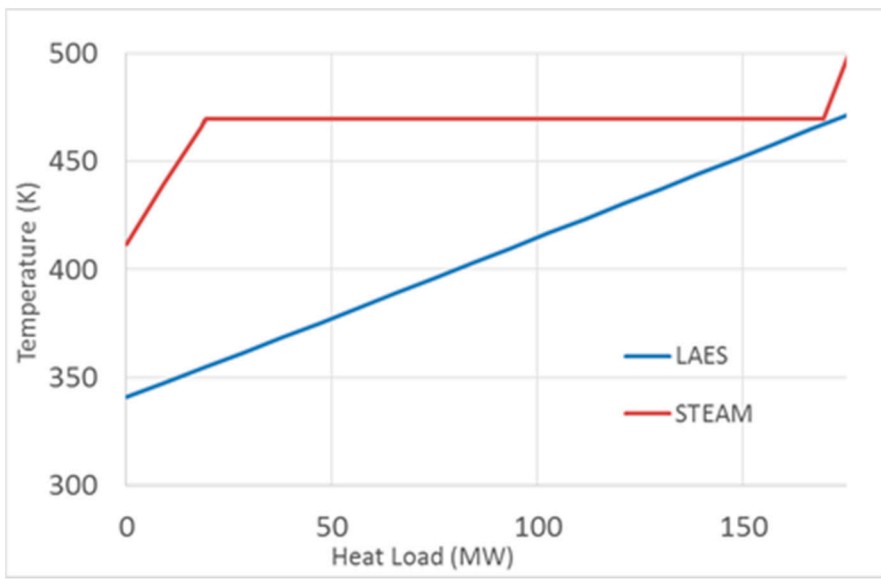

**Figure 6.** Temperature profile of heat exchanger 4.

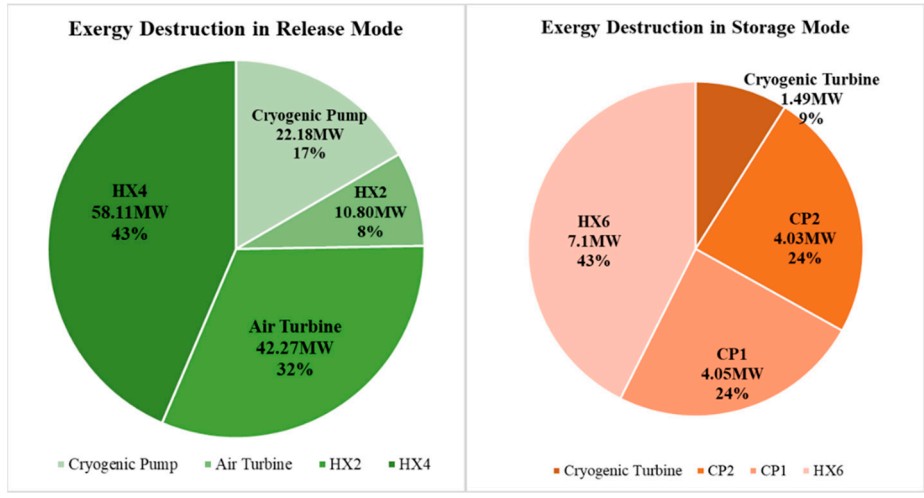

**Figure 7.** Exergy destruction in release mode (**Left**) and in storage mode (**Right**).

**Table 9.** Performance summary for APR1400 integrated LAES.

| Component | Power (MW) | Exergy LOSS (MW) |
|---|---|---|
| Energy storage mode | | |
| Compressor1 | 40.11 | 4.05 |
| Compressor2 | 39.87 | 4.03 |
| Cryogenic turbine | 3.30 | 1.64 |
| Energy release mode | | |
| Cryogenic pump | 21.38 | 22.18 |
| Air turbines | 580.41 | 42.27 |
| Net Power Consumption in Storage Mode (MW), $W_{ES}$ | | 76.68 |
| Net Power Output in Release Mode (MW), $W_{ER}$ | | 599.04 |
| Thermal Power from NPP (MW), $W_{NPP}$ | | 191.21 |
| Round-trip Efficiency (%), $\eta_{RTE}$ | | 59.96 |

*3.4. Pressure Drop and Ambient Temperature Effects*

In any type of hydrodynamic component and pipe, the pressure of fluid is decreased by frictional pressure drop while traveling in the components and pipelines. In this study, the pressure drop is modeled as a fractional pressure drop of the system pressure while ignoring the shape and flow path of air going through the system. The pressure drop changes the properties of fluid; therefore, the output and consumption power of each component will change. The increase of pressure drop lowers the round-trip efficiency, shown in the top of Figure 8. This is because the liquid air yield is substantially reduced in the separator, which leads to a decrease in overall flow rate of the LAES system during release mode as shown in the bottom of Figure 8. Consequently, the reduced mass flow rate of air in the heat exchanger 4 results in the decreased steam mass flow rate from the steam cycle, and thus, the delivered thermal power is reduced to have an increasing effect on the round-trip efficiency. However, net output power from the air turbines reduces due to lowered expansion ratio, and overall, the round-trip efficiency decreases. It is thus necessary to minimize the number of components that make up the system in order to prevent round-trip efficiency degradation caused by pressure drop in the actual system.

The sensitivity of the overall system to the change in ambient temperature is also studied. As the LAES system is an open system that inhales air from the outside of the system to operate, there is a significant impact on the operation with respect to the ambient condition. The ambient temperature was set to 288K in the above calculations. However, the ambient temperature can increase or decrease due to seasonal change or the weather conditions in the installed location. Since it is trivial that a decrease of the ambient temperature will be helpful to increase the round-trip efficiency, as cold air will consume less energy to be liquefied, only the increase of ambient temperature case is presented in this study.

The rise in ambient temperature decreases the round-trip efficiency of the system because of the increased energy consumption in storage mode. Due to the nature of the LAES using the air as an energy storage medium, the higher the temperature of the ambient air, the more energy is consumed to cool the air, as shown in the top of Figure 9. The increase in ambient air temperature also increases the temperature of the No. 5 hot side inlet of heat exchanger 6. This increases the temperatures of outlet No. 6 and No. 9 from heat exchanger 6, which leads to a decrease in the liquid air yield. Moreover, the reduction of the liquid air yield reduces the mass flow rate in the system during the release mode and consequently reduces the generated power of the release mode. This trend is shown in the bottom of Figure 9. The reduction of power output leads to a decrease in round-trip efficiency. The round-trip efficiency is further deteriorated by the net power consumption increase during the storage mode, which is shown in the top of Figure 9. This consequently reduces the round-trip efficiency as the ambient temperature increases.

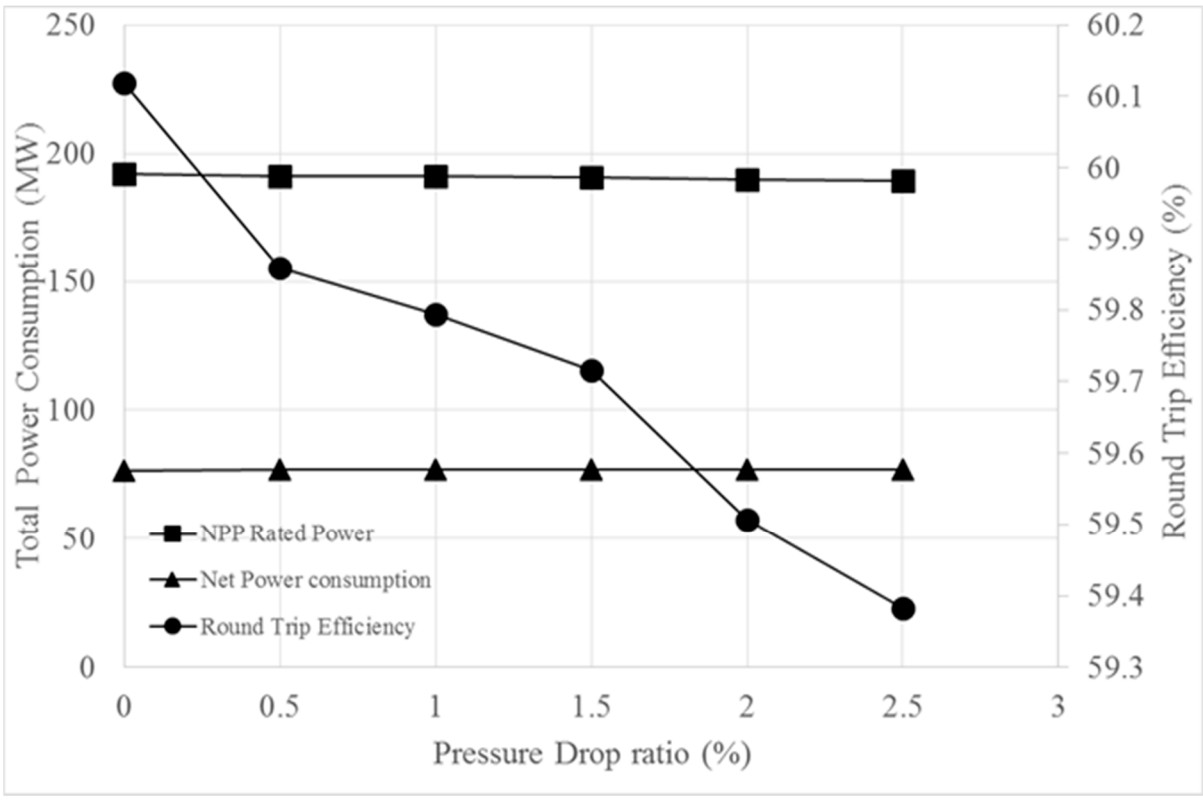

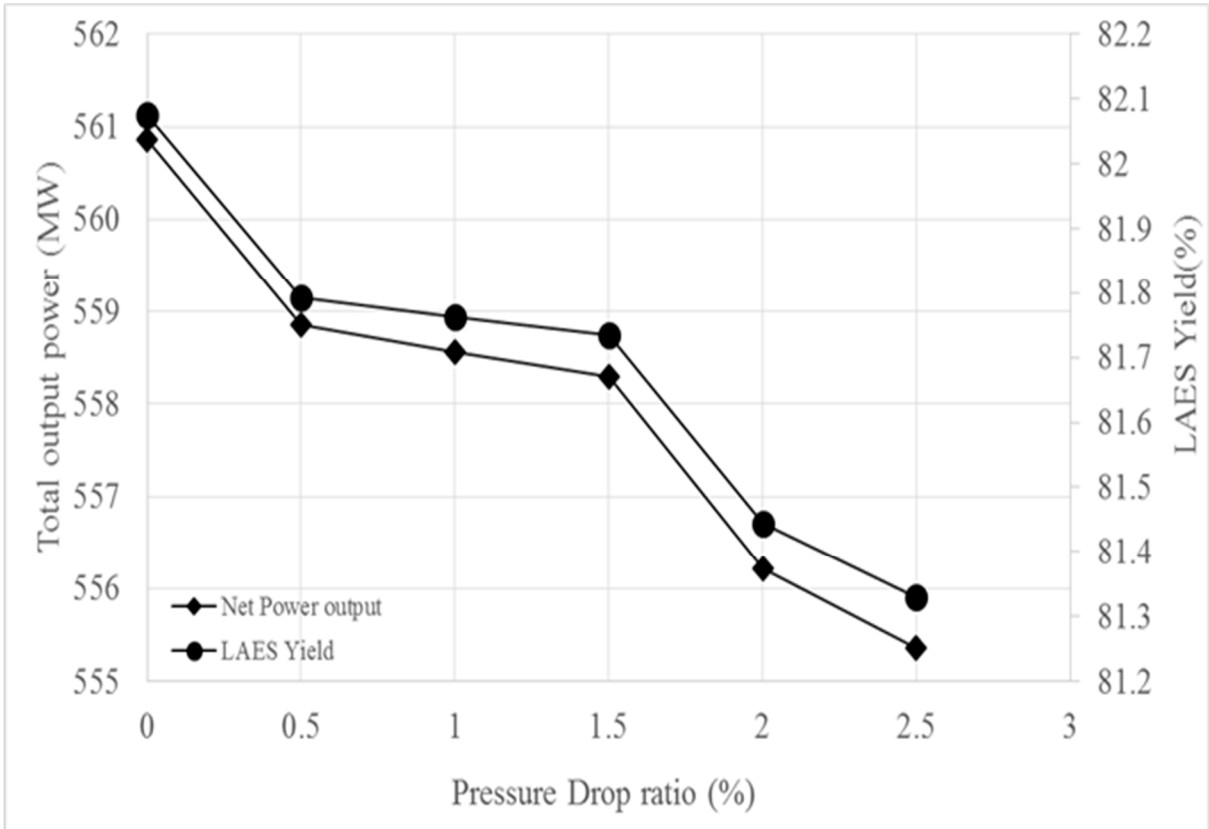

**Figure 8.** Pressure drop, power consumption, and round trip efficiency (**Top**), LAES Yield (**Bottom**).

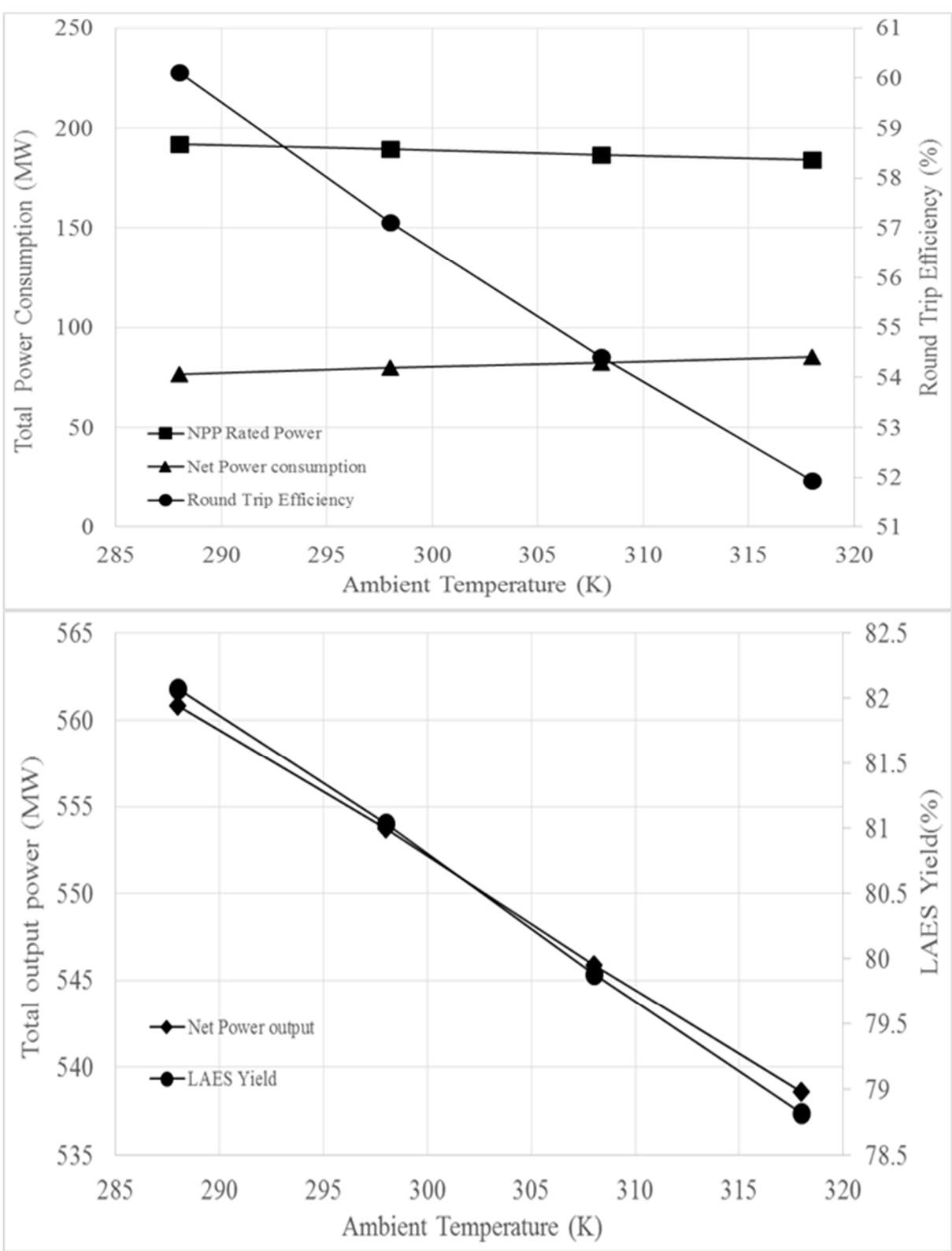

**Figure 9.** Ambient Temperature, power consumption and round-trip efficiency (**Top**), LAES Yield (**Bottom**).

## 4. Conclusions

A model was reproduced from the previously suggested system concept in the previous work [15], and the optimization for realistic operating conditions was completed in this project. An in-house code was developed and utilized to model the liquid air energy

storage (LAES) system combined with a nuclear power plant (NPP). The minor differences in the reference comparison, rooting in the cryogenic components having notable range of differences in the mixture properties, were re-evaluated in this study more realistically.

The optimization was focused on the conditions targeting the realistic design of the system. The pinch of heat exchanger was limited to 5K which is the common pinch used in the industry. In the 5K pinch case, the power consumption in the storage mode increased, and the power output in the release mode decreased. Moreover, in the actual operation, pressure drop and changes in the ambient temperature are unavoidable. After reflecting these realistic considerations, the round-trip efficiency of the ideal value for the system is 67.6%, which is lower than reported value in the reference [15]. For the 2K pinch case, it is slightly higher at 68.29%.

Moreover, the round-trip efficiency decreases to 59.96% under the realistic operation conditions from the APR1400 nuclear power plant. Considering the technical and safety issues to bypass a high-pressure turbine from the nuclear power plant, this result is an acceptable alternative to integrate the LAES and the nuclear power plant. Given the pressure drop and changes in the ambient temperature, the actual round-trip efficiency of the LAES nuclear power plant system is expected to have difficulty exceeding 60% efficiency. However, the combination of these two systems presents a new possibility for the original non-existing load following operations. In addition, the LAES-NPP system can expect higher economic value because some of the energy to be discarded is stored in a usable form, whereas conventional load following mode operates as a reduction of reactor power which reduces the capacity factor.

The following observations were made in the analysis when the realistic design conditions were applied:

1.  Pinch temperature of heat exchangers—storage mode power consumption increased and release mode power output decreased
2.  Realistic NPP steam conditions—thermal power from NPP decreased, LAES yield decreased, and release mode power output decreased
3.  Pressure drop—round-trip efficiency is slightly decreased
4.  Ambient temperature—round-trip efficiency decreased substantially

In order to increase the efficiency of the integrated systems, the LAES system must be linked under conditions that do not affect the normal operation of the plant. It also requires a design that can efficiently transfer the thermal power of the nuclear power plant to LAES under those conditions. While expanding the heat exchange area, length limit of thermal components is required to minimize pressure drop even under such design conditions. Finally, external factors should also be considered as the LAES-NPP integrated system is sensitive to ambient temperature. These realistic considerations will facilitate the deployment of the LAES-NPP system in the near future.

**Author Contributions:** Conceptualization, S.-H.S. and J.-Y.H.; methodology, S.-H.S. and J.-Y.H.; software, S.-H.S. and J.-Y.H.; validation, S.-H.S., J.-Y.H. and J.-I.L.; formal analysis, S.-H.S.; investigation, S.-H.S.; resources, S.-H.S.; data curation, S.-H.S.; writing—original draft preparation, S.-H.S.; writing—review and editing, S.-H.S., J.-Y.H. and J.-I.L.; visualization, S.-H.S.; supervision, J.-I.L.; project administration, J.-I.L. All authors have read and agreed to the published version of the manuscript.

**Funding:** This research received no external funding.

**Institutional Review Board Statement:** Not applicable.

**Informed Consent Statement:** Not applicable.

**Data Availability Statement:** The data used in this study are openly available in the reference [15,17].

**Conflicts of Interest:** The authors declare no conflict of interest.

### Nomenclature

| | |
|---|---|
| CCD | Closed Cycle Design |
| CP | Compressor |
| $\epsilon$ | Effectiveness |
| ES | Energy Storage |
| ER | Energy Release |
| T | Operation time of system |
| LAES | Liquefied Air Energy Storage |
| NPP | Nuclear Power Plant |
| H | Enthalpy |
| $\dot{m}$ | Mass Flow Rate |
| H | Efficiency of component |
| $\eta_{RTE}$ | Round-trip Efficiency |
| E | Exergy per unit mass |
| Q | Heat |
| W | Work |
| X | Exergy Destruction of component |
| TB | Turbine |
| HX | Heat Exchanger |
| Y | LAES Yield |

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
