# Peer review of "Design Considerations for the Liquid Air Energy Storage System Integrated to Nuclear Steam Cycle"

_applsci, doi:10.3390/app11188484_

Round 1

Reviewer 1 Report

This paper studies the applicability of a liquid air energy storage system in combination with nuclear power plants, to reduce the impact of load fluctuation and improve the overall efficiency of the plant. The topic is current and very interesting, however the paper has serious concerns especially related with the format and readability. Specifically, I have the following concerns:

  • Personally, I do not understand what the lines 8-9 mean.
  • Please revise the English and format throughout the paper, for example in line 19 where the format is not consistent with the rest of the manuscript.
  • The authors should comment the issues of nuclear and other conventional generation plants on the face of grid outages and how small renewable generators could help local consumers to reduce their impact. In this regard, I suggest the reference [A].

[A]         M. Tostado-Véliz, D. Icaza-Alvarez, F. Jurado. A novel methodology for optimal sizing photovoltaic-battery systems in smart homes considering grid outages and demand response. Renewable Energy 2021; 170: 884-896. https://doi.org/10.1016/j.renene.2021.02.006

  • The reference format is not consistent throughout the paper. One example can be found in line 110, where the authors use a reference format which is not consistent with that used in MDPI journals.
  • I am concerned with the resolution of Fig. 3. It seems clear that authors have copied-paste this figure from reference [16], which has led to resolution issues. In my opinion. The authors could draw the figure by themselves and avoid such issues. In such case, the authors should clarify that the figure is adapted from [16].
  • Resolution of some expressions like (1) and (2) is quite poor. This is due to authors have inserted these equations as images. I suggest using some math editor like Mathtype or the MS Word equations editor.
  • Please numerate all the sections and subsections.
  • In general, format of the manuscript should be notably improved. It is occasionally difficult to read.
  • In the results section some tables are not introduced in the text. As for example the Table 3. In this sense, it is difficult for me revising these tables without knowing what they actually mean.
  • The authors should compare (or at least provide some comments about the differences) the developed method with the approach proposed in ‘A Novel Methodology for Comprehensive Planning of Battery Storage Systems’ for planning storage systems. Could the suggested methodology be applied to the problem concerned in the paper? Please include some comments in the revised manuscript.

Thanks to the authors for your effort and time.

Author Response

First of all, thank you for your precious time for the review. All the comments have been of great help in improving the quality of the paper. While responding to the comments the quality of the paper have been improved substantially, and the improvement of the paper greatly owes to the reviewers’ time and effort for sharing their views. The authors wish the revision can satisfy the reviewers’ standards. Thank you

Author Response

(green highlight: authors’ responses, yellow highlight: revised text of the manuscript)

First of all, thank you for your precious time for the review. All the comments have been of great help in improving the quality of the paper. While responding to the comments the quality of the paper have been improved substantially, and the improvement of the paper greatly owes to the reviewers’ time and effort for sharing their views. The authors wish the revision can satisfy the reviewers’ standards. Thank you

Reviewer 3 Report

Dear Author, The comments are given in the attached file.

Author Response

(The authors gave the same response as above.)

Round 2

Reviewer 1 Report

Dear authors,

All my concerns have been properly addressed.

I have not further comments.

Congratulations!

Author Response

Thank you again for taking your valuable time in the review. All the comments had been of great help in improving the quality of the paper. While responding to the comments the quality of the paper had been improved substantially, and the improvement of the paper greatly owed to the reviewers’ time and effort for sharing their views. Thank you.

Author Response

Thank you again for taking your valuable time in the review. All the comments have been of great help in improving the quality of the paper. While responding to the comments the quality of the paper has been improved substantially, and the improvement of the paper greatly owes to the reviewers’ time and effort for sharing their views. The authors wish the revision can satisfy the reviewers’ standards. Thank you. 

Reviewer 3 Report

  • Correct page layout, in particular left margin
  • Figure 1 is improved, but component names can be enlarged
  • Figure 2: Header below Figure
  • Line 136: ….with the exception of heat exchanger… which one?
  • Table 2: change header to: “Modification of the LAES calculation based on the coupling with APR1400”. Highlight differences between Table 1 and 2 in bold print.
  • Line 175: write: In paper [16]….
  • [1]: I think wx,I has to be written in capital letter because it is not mass specific.
  • 6 and 8: Capital letters for W all over the equations in order not to be confused with mass specific symbols
  • Table 4 is placed in the wrong section. Place it to 3.1
  • Section 3.3: Regarding Fig.1, I have problems to understand why there is a lower pressure of steam at the inlet of HX4 (No. 33 in Table 8) if a flow bypass is taken from the steam turbine inlet. I assume, the steam parameters (temperature, pressure) at the turbine´s entrance of the NPP remain the same independent from coupling to LAES. Could you clarify?
  • One question arises: When changing air temperature, pressure drop and steam flow, how does this effect pinch in HX6?
  • Nomenclature: W, resp. w, Q and index C are missing

Author Response

(The authors gave the same response as above.)
